# Potential Therapeutic Exploitation of G Protein-Coupled Receptor 120 (GPR120/FFAR4) Signaling in Obesity-Related Metabolic Disorders

**DOI:** 10.3390/ijms26062501

**Published:** 2025-03-11

**Authors:** Dariusz Szukiewicz

**Affiliations:** Department of Biophysics, Physiology & Pathophysiology, Faculty of Health Sciences, Medical University of Warsaw, 02-004 Warsaw, Poland; dariusz.szukiewicz@wum.edu.pl

**Keywords:** G protein-coupled receptor 120, free fatty acid receptor 4, obesity, nonalcoholic fatty liver disease, anti-inflammatory signaling, metabolic effects, adipose tissue, obesity treatment

## Abstract

The increasing prevalence of overweight and obesity not only in adults but also among children and adolescents has become one of the most alarming health problems worldwide. Metabolic disorders accompanying fat accumulation during pathological weight gain induce chronic low-grade inflammation, which, in a vicious cycle, increases the immune response through pro-inflammatory changes in the cytokine (adipokine) profile. Obesity decreases life expectancy, largely because obese individuals are at an increased risk of many medical complications, often referred to as metabolic syndrome, which refers to the co-occurrence of insulin resistance (IR), impaired glucose tolerance, type 2 diabetes (T2D), atherogenic dyslipidemia, hypertension, and premature ischemic heart disease. Metabotropic G protein-coupled receptors (GPCRs) constitute the most numerous and diverse group of cell surface transmembrane receptors in eukaryotes. Among the GPCRs, researchers are focusing on the connection of G protein-coupled receptor 120 (GPR120), also known as free fatty acid receptor 4 (FFAR4), with signaling pathways regulating the inflammatory response and insulin sensitivity. This review presents the current state of knowledge concerning the involvement of GPR120 in anti-inflammatory and metabolic signaling. Since both inflammation in adipose tissue and insulin resistance are key problems in obesity, there is a rationale for the development of novel, GPR120-based therapies for overweight and obese individuals. The main problems associated with introducing this type of treatment into clinical practice are also discussed.

## 1. Introduction

Abnormal chemical reactions occurring in the body may lead to a disruption of normal metabolic processes [1]. Metabolic disorders adversely affect the processing and distribution of nutrients in the body, including macronutrients such as proteins, fats, and carbohydrates, as well as micronutrients (mostly vitamins and minerals), which are equally important but are consumed in very small amounts [2,3,4]. Metabolic disorders in humans are an increasingly recognized health problem in modern civilization, and in developed countries, the disease of obesity has come to the forefront [5]. Over the past 25 to 30 years, the incidence of overweight and obesity has increased at an alarming rate, becoming a pandemic [6,7,8]. According to data from the World Obesity Organization, between 1975 and 2022, the incidence of obesity among women almost tripled (6.6% to 18.5%), and among men, it increased fourfold (3% to 14.0%) [9]. This increased incidence of obesity in adults is accompanied by a growing problem of obesity in children and adolescents. Compared with 2000–2011, in 2012–2023, a 1.5-fold increase in the incidence of obesity occurred among children and adolescents worldwide [10].

Excessive fat accumulation, which causes health risks, is reflected in the increased body mass index (BMI) values that define overweight (BMI > 25 kg/m^2^) and generalized obesity (BMI > 30 kg/m^2^), among which extreme obesity (BMI > 40 kg/m^2^) and central obesity can also be distinguished [11]. Chronic pathological weight gain in obese individuals is considered multifactorial with strong genetic and epigenetic components [12,13]. Owing to the genetic background, numerous cultural and environmental factors may affect mental characteristics and behavior, ultimately disrupting hormonal and metabolic homeostasis toward fat accumulation [14,15].

Obesity, in particular, is associated with a reduced life expectancy, largely because obese individuals are at an increased risk of many medical complications [16]. Well-known diseases comorbid with obesity include type 2 diabetes mellitus (T2D), hypertension, heart disease, stroke, metabolic syndrome (also called insulin resistance syndrome), nonalcoholic fatty liver disease (NAFLD), chronic kidney disease, some cancers (e.g., colorectal cancer, liver cancer, breast cancer, and prostate cancer), breathing problems, gout, osteoarthritis, sexual dysfunction, fertility disorders, and mental health problems [16,17,18,19,20,21,22,23].

A phenomenon that has not been appreciated until recently in the course of obesity is the persistence of a low-grade inflammatory response in fatty tissues [24,25]. In particular, excess adipose tissue (AT) macrophages present in the visceral AT of individuals with central obesity induce chronic inflammation, exacerbating insulin resistance (IR) by interfering with insulin signaling via insulin receptors [26]. Moreover, glucose intolerance in IR, which manifests as hyperglycemia, additionally stimulates the primary defensive inflammatory response. The numerous pro-inflammatory cytokines released in individuals with these conditions pose a real risk of deterioration of organ function [27,28].

Progress in understanding the pathophysiology of obesity therefore prompts the search for therapeutic effects on metabolic pathways through the identification of key receptors that mediate anti-inflammatory responses. G protein-coupled receptors (GPCRs), also known as seven transmembrane domain (7TM) receptors, are the largest and most diverse group of integral membrane (cell surface) metabotropic receptors in eukaryotes [29,30]. Among the GPCRs that are relatively widely expressed in many tissues of various human and rodent organs and adipose tissue, G protein-coupled receptor 120 (GPR120), also known as free fatty acid receptor (FFAR4), is of interest [31,32].

This review presents the current state of knowledge concerning the involvement of GPR120 in anti-inflammatory and metabolic signaling, which may provide a rationale for the development of novel, GPR120-based therapies for overweight and obese individuals. The main problems associated with introducing this type of treatment into clinical practice are also discussed.

## 2. G Protein-Coupled Receptor (GPCR) Family

The expression of GPCRs was described for the first time by an American molecular biologist and physician, Robert J. Lefkowitz, in the 1970s. Detailed studies of the structures and roles of GPCRs were then initiated by his colleague Brian K. Kobilka. For their achievements in the field of GPCR research, which were “crucial for understanding how G protein-coupled receptors function”, both outstanding researchers were awarded the Nobel Prize in Chemistry in 2012 [33].

The characteristic 7TM structure of a GPCR with six loops causes the detection of bound molecular ligands on the outer surface of the cell to activate intracellular responses. The three extracellular loops interact with ligands on the N-terminal extracellular side of the GPCR, whereas the three intracellular loops interact with G proteins on the C-terminal (intracellular) side of the receptor [29,30] (Figure 1).

G proteins form heterotrimeric complexes that consist of the α (39–52 kDa), β (35–36 kDa), and γ (7–10 kDa) subunits. G protein classes are defined based on the sequence and function of their Gα subunits, which are categorized into several subtypes in mammals: Gαi, Gαs, Gα12/13, Gαq, and transducin (GαTr). The observed differences between these classes result mainly from the fact that Gαi and Gαs regulate the activity of adenylyl cyclase, whereas Gα12/13, Gαq, and GαTr activate small GTPase family proteins and phospholipase C-beta (PLC-β), respectively [35,36].

Conformational changes in the GPCR after ligand binding activate the guanine nucleotide exchange factor (GEF) on the intracellular side, thus activating monomeric GTPase by stimulating the release of guanosine diphosphate (GDP), which then binds to guanosine triphosphate (GTP) [34]. Activation of the G protein, which has inherent GTPase activity, involves the exchange of GDP for GTP in the Gα subunit, which leads to the reversible dissociation of the Gα subunit from the tightly connected Gβ and Gγ subunits that form a separate functional unit. Therefore, the G protein complex functions through a cycle of dissociation of the Gα and Gβ/Gγ subunits, and it functions as a molecular binary switch [35,37]. Intracellular downstream signaling via Gα-GTP and Gβγ, as well as the targeting of functional proteins after GPCR activation, directly depend on the Gα subunit type (Gαi, Gαs, Gα12/13, or Gαq). The G protein remains active until the GTP bound to the Gα subunit is hydrolyzed to GDP [38]. The intracellular signaling cascades activated by GPCRs that lead to the cyclic adenosine 3,5-monophosphate (cAMP) response, calcium mobilization, or the phosphorylation of extracellular signal-regulated protein kinases 1/2 (pERK1/2) have been considered to be remarkably complex, similar to the biological processes they regulate [39,40,41].

Approximately 850 different genes encoding GPCR proteins have been detected in the human genome, accounting for approximately 4% of all human coding genes [42]. Thus, GPCRs, which constitute the most abundant superfamily of integral membrane proteins, represent the largest protein family encoded in the human genome [41,43,44].

Given the crucial role of cell membranes in cell life, the finding that the spectrum of signaling through GPCRs is extremely broad is not surprising, covering most physiological responses to neurotransmitters, hormones, autocrine and paracrine signaling molecules, and environmental stimuli [45,46,47]. Therefore, the group of endogenous GPCR ligands is very diverse, ranging from photons, ions, biogenic amines, amino acids, odorants, pheromones, carbohydrates, peptides, lipids, eicosanoids, chemokines, and neurotransmitters to proteins and hormones [41,48]. Nevertheless, for approximately 30% of nonolfactory human GPCRs, their natural ligands have not been identified, which is why they remain orphan receptors [49,50].

GPCR signaling disorders are associated with the occurrence of many different diseases, including but not limited to type 2 diabetes (T2D), hypo- and hyperthyroidism, obesity, Alzheimer’s disease (AD), cancer, pain, cardiac problems, asthma, depression, and fertility issues [35,51,52,53,54,55]. For example, disease-causing mutations have been reported in at least 55 GPCRs for more than 66 inherited monogenic diseases in humans [56,57].

Notably, of the 826 GPCR family proteins identified thus far in humans, according to various estimates, approximately 350 to 400 nonolfactory proteins are considered druggable, which, in 165 cases, has been validated in the development of targeted drugs [41,58,59]. Nearly 40% of all U.S. Food and Drug Administration (FDA)-approved drugs currently target GPCRs to achieve therapeutic goals, such as treating hypertension, chronic pain, diabetes and other endocrinopathies, cancer, or neurological diseases [41,59,60,61,62].

### 2.1. Classification of GPCRs

Many classifications have been developed for the superfamily of GPCRs, but in practice, the two most commonly used are as follows:-The classical A–F system, in which GPCRs are grouped into six classes based on sequence homology and functional similarity

A (or 1), rhodopsin-like receptors;

B (or 2), secretin receptor family;

C (or 3), metabotropic glutamate/pheromone receptors;

D (or 4), fungal mating pheromone receptors;

E (or 5), cyclic AMP receptors;

F (or 6), frizzled/smoothened receptors;

-A newer alternative classification proposed for vertebrates, known by its acronym GRAFS, which stands for glutamate, rhodopsin, adhesion, Frizzled/Taste2, and Secretin. The GRAFS system corresponds to classical classes C, A, B2 (Secretin receptor family, long N-terminal), F, and B1 + 3 (other secretins) [63,64,65].

The most frequently represented class among GPCRs is class A (rhodopsin-like), which includes approximately 83% of GPCR genes and translates into 241 nonolfactory receptors of a total of 701 [66]. GPR120/FFAR4 belongs to this large family of rhodopsin-like receptors [67].

### 2.2. G Protein-Coupled Receptor 120 (GPR120)

The rhodopsin-like class of GPCRs includes, among others, receptors whose activators are fatty acids of various chain lengths. For example, short-chain fatty acids (SCFAs) bind to GPR41 and GPR43 [68,69], medium-chain fatty acids (MCFAs) bind to GPR84 [70], and LCFAs bind to GPR40 and GPR120/FFAR4 [71].

GPR120/FFAR4 is a sensor that detects saturated and unsaturated long-chain fatty acids (LCFAs) with 14–18 and 16–22 carbons, respectively. Therefore, GPR120 is a functional receptor for omega-3 fatty acids (Ω-3 FAs) and the related plant-derived molecule alpha-linolenic acid (ALA) [72].

When the results of studies on GPR120 conducted in animals are applied to the human body, important differences should be taken into account. For example, the amino acid sequences of mouse (BC053698) and human (BC101175) GPR120 exhibit an 82% match [73].

Compared with human GPR120, murine GPR120 has a stronger response to low doses of agonists, and dose–response analysis revealed a lower expression level of murine GPR120. Interestingly, when comparing human and mouse receptors, no differences in the half maximal effective concentration (EC50) were observed between the two GPR120s [73,74].

Only one GPR120 variant, composed of 361 amino acids, has been detected in rodents and cynomolgus monkeys, while the human GPR120 gene encodes two protein isoforms [75,76]. The chromosomal position of GPR120 in humans is 10q23.33, and the two splice variants that occur are known as short GPR120 (GPR120S) and long GPR120 (GPR120L), containing 361 and 377 amino acids, respectively [77]. The insertion of 16 amino acids between positions 231 and 247 in the third intracellular loop (ICL3) in GPR120L prevents G protein-dependent intracellular calcium and dynamic mass redistribution in the absence of a functional connection with the adaptor protein β-arrestin [76,78]. Human GPR120 appears to correlate more closely with the long form of this receptor, which is one of the first examples of a native β-arrestin-biased receptor [75]. The domain within ICL3 is crucial for binding to G proteins and interacting with β-arrestin during GPR120 activation. In the basal state, GPR120L is phosphorylated at a lower level than GPR120S is [79].

**Determinants and localization of GPR120 synthesis**. The rate of GPR120 receptor synthesis is influenced by genetic and epigenetic (environmental) factors [80]. Studies of the Danish population have shown that the gene polymorphism leading to the p.R270H variant is associated with a 70% decrease in GPR120 expression compared with that in people without this mutation [81]. The expression level of GPR120 also varies depending on the contents of its natural ligands [eicosapentaenoic acid (EPA), docosahexaenoic acid (DHA), and ALA] in the diet and the location (cell type) in the body [82]. Compared with control rats fed soybean oil, rats fed a diet with an increased content of fish and/or linseed oil presented significantly higher GPR120 expression in the cells of the large intestine [83]. Studies in rats have also shown that the dietary administration of unsaturated Ω-3 FAs leads to an increase in GPR120 expression in the central nervous system (CNS) [84]. In rats, a high-fat diet led to increased expression of GPR120 in the cardiac muscle and extensor digitorum longus (skeletal muscle), whereas no changes in GPR120 expression were detected in the liver or soleus muscle (skeletal muscle) [85,86]. The effect of a high-fat diet on tissue-specific changes in GPR120 expression was also observed in mice. For example, GPR120 mRNA expression was already high at baseline in four different adipose tissue depots (subcutaneous, perimesenteric, epididymal, and periuterine) and then became upregulated after the consumption of a high-fat diet. The expression of the GPR120 mRNA was higher in adipocytes than in stromal–vascular (S-V) cells. However, in mice, a high-fat diet had no effect on GPR120 expression in perirenal adipose tissue, the nucleus accumbens (NAcs), or the mid-basal part of the hypothalamus [86,87].

In situ hybridization and immunohistochemistry revealed GPR120 expression in mouse gonadotropes of the anterior pituitary gland (adenohypophysis) but not in thyrotropes, somatotropes, lactotropes, corticotropes, melanotropes, or the posterior pituitary gland [88]. Moreover, 24 h of fasting induced the upregulation of GPR120 mRNA expression in the pituitary gland, which may indicate that GPR120 in pituitary gonadotropes is involved in controlling gonadotropin synthesis and secretion [88,89,90].

**GPR120 signaling**. Importantly, the activation of GPR120 and the subsequent metabolic effects related to the interaction of the receptor with the Gα protein via activated extracellular signal-regulated kinase (ERK) cascades and increases in the intracellular Ca^2+^ concentration seem to be potentially beneficial in treating obesity by promoting anti-inflammatory and insulin-sensitizing effects [91,92,93]. The β-arrestin2 (ARRβ2)–transforming growth factor-beta (TGFβ)-activated kinase 1 (TAK1)-binding protein 1 (TAB1) complex and the inhibition of the TAK1–c-Jun N-terminal kinase (JNK)–nuclear factor kappa-light-chain-enhancer of activated B cells (NF-κB) pathway play key roles in the anti-inflammatory mechanism induced by GPR120 stimulation [82,94,95]. The main signaling pathways associated with GPR120/FFAR4 stimulation are presented in detail in Figure 2.

## 3. Inflammatory Background of Metabolic Diseases

### 3.1. Inflammatory Response in Obesity

The appearance of a chronic inflammatory component in obesity is a direct derivative of IR. Under physiological conditions, insulin not only provides glycemic control but also has anti-inflammatory effects [113]. Under the influence of insulin, the nucleotide-binding oligomerization domain (NOD), leucine-rich repeat (LRR)-containing protein (NLR) family member 3 (NLRP3) inflammasome, which is extremely important for the innate immune system, is downregulated [114]. This immunomodulation significantly decreases the activation of caspase-1 and the secretion of the pro-inflammatory cytokines IL-1β/IL-18 in response to cell damage and bacterial infection. Accordingly, in vitro and in vivo studies have shown that the alleviation of inflammation symptoms after a pretreatment with insulin is accompanied by the significant inhibition of nuclear factor kappa-light-chain-enhancer of activated B cell (NF-κB) activation along with a subsequent decrease in the expression of pro-inflammatory cytokines [115,116]. Chronic IR, which deprives insulin of its effective hypoglycemic and anti-inflammatory effects, leads to the persistence of low-grade inflammation, which is typically observed in obese individuals [117]. The paradox is that this process usually happens despite hyperinsulinemia. Moreover, a vicious cycle occurs in which excess visceral adipose tissue (AT) in obese individuals initiates a chronic low-grade inflammatory background that interferes with insulin signaling via insulin receptors (INSRs), thereby intensifying IR [28]. Moreover, hyperglycemia, which is a derivative of IR itself (without additional metabolic factors), stimulates a primarily defensive inflammatory response, which is associated with the release of numerous inflammatory cytokines, including interferon gamma (IFN-γ), tumor necrosis factor alpha (TNFα), interleukins (IL-1β, IL-6, and IL-8), and interleukin 1 receptor antagonist (IL-1RA), and poses a real threat to the proper functioning of organs [118]. This response is accompanied by a change in the profile and ratio of classically activated (M1) and alternatively activated (M2) macrophages, which are essential regulators of inflammation, in a pro-inflammatory direction [119]. This chronic state of inflammation, termed “meta-inflammation” and mediated by macrophages in AT (and also in the colon, liver, and muscles), involves interactions with components of the adaptive immune response through complex and not fully understood mechanisms [120,121,122]. Therefore, the disturbed balance of Th1/Th2 cytokine profiles resulting from the shift from Th2 cells to Th1 cells and additionally toward Th17 cells and cytotoxic T lymphocytes (CTLs) is not simply the result of the direct impact of IR on lymphocyte subpopulations [123].

In particular, low-grade systemic inflammation and increased reactive oxygen species (ROS) production (oxidative stress) can be observed in individuals with visceral obesity [124]. Visceral obesity, which is characterized by an increased mass of AT surrounding intra-abdominal organs, is also referred to as abdominal or central obesity [125]. Visceral obesity itself is an independent component of metabolic syndrome, the cluster of metabolic factors that also includes high blood pressure, impaired fasting glucose levels, high triglyceride levels, and low high-density lipoprotein (HDL) cholesterol levels [126]. Metabolic syndrome greatly increases the risks of developing diabetes, heart disease, stroke, or all three, and visceral fat accumulation appears to be an accurate predictor of metabolic syndrome [127,128,129]. In other words, visceral obesity may be a partial marker of a dysmetabolic condition that predisposes individuals to the occurrence of diabetogenic, atherogenic, prothrombotic, and pro-inflammatory events and their consequences (complications) [117,130,131]. Moreover, the disruption of numerous endocrine, metabolic, and immunological functions of AT, which is especially pronounced in individuals with visceral obesity, increases the risk of cancer development, including breast cancer, esophageal adenocarcinoma, and colorectal adenocarcinoma [132,133,134].

Notably, the inflammatory process accompanying excess AT is not limited to AT but is generalized (systemic) [135]. A large body of evidence indicates that the regional distribution of body fat, rather than overall obesity, is linked to systemic inflammation. Thus, increased visceral fat promotes generalized inflammation [136,137]. The inflammatory response, especially an increased tendency toward macrophage infiltration, spreads to other tissues, and modifications of the cell environment caused by excess FFAs and increases in the concentrations of growth factors and pro-inflammatory cytokines have been observed in the liver, pancreas, heart, and skeletal muscles [138,139]. For example, among the activities of enzymes and other molecules produced by visceral fat and released from macrophages, the adipocytokine visfatin/nicotinamide phosphoribosyl transferase (NAMPT) has been linked to several inflammatory conditions outside AT [117,140,141]. Unlike the intracellular form of Visfatin/NAMPT (iNAMPT), which plays a regulatory role in NAD+ biosynthesis and thereby affects many NAD-dependent proteins [e.g., sirtuins, poly ADP–ribose polymerases (PARPs), mono ADP–ribosyl transferases (MARTs), and ADP–ribosyl cyclases (CD38/157)], the extracellular form (eNAMPT) can upregulate pro-inflammatory cytokines and matrix metalloproteinases (MMPs) in various types of cells, with potentially important effects on glucose metabolism and atherosclerosis [142,143]. Serum visfatin/NAMPT levels are significantly higher in obese subjects than in control subjects [144].

The multifactorial pathomechanism of the development of IR and the inflammatory response in individuals with visceral obesity is summarized in Figure 3.

### 3.2. Anti-Inflammatory and Metabolic Effects of GPR120 Signaling in the Context of Overweight and Obesity

In overweight and obese individuals, significant homeostasis disorders are observed at the level of adipose tissue, cells lining the digestive tract that produce gastrointestinal hormones, and the endocrine function of the pancreas [99,160,161,162]. The vast majority of cells present in all these locations express GPR120. Therefore, it is important to determine whether obesity-induced changes in the functional GPR120 signaling pathways that may cause metabolism-induced inflammation (meta-inflammation) are justified [97,163,164]. This cause-and-effect relationship takes a new dimension after considering reports that the GPR120 genetic variant p.Arg270His is detected more often in obese people and that this genetic variant is functionally associated with obesity in humans [165].

In the context of such comprehensive disturbances of GPR120 signaling in overweight and obesity, the importance of proper nutrition should be taken into account [166]. In the diet, in addition to individually selected caloric restrictions in order to reduce body weight, attention should be paid to food quality, including healthy (monounsaturated and polyunsaturated) fats and sufficient micronutrient content [2,3,4,167]. For example, coconut oil products containing lauric acid, whose weight-controlling effects are well known, may have a beneficial effect on metabolism by providing effective ligands for GPR84 (MCFAs), GPR40 (LCFAs), and GPR120 (LCFAs) [168,169,170]. Additionally, actions aimed at reducing the environmental contamination of food with various toxins with a potential impact on GPRs signaling may be important [1,171,172].

#### 3.2.1. GPR120 in Adipose Tissue

The importance of AT in the body extends far beyond its role in maintaining energy homeostasis through energy storage and release, insulation from cold and heat, or cushioning around soft organs. Adipocytes represent a unique population of specialized cells involved in endocrine, nervous, and immune functions, which translate into the regulation of glucose and cholesterol levels; the maintenance of insulin sensitivity; the metabolism of sex hormones; the regulation of hunger and satiety; and, in cooperation with resident immune cells, the regulation of the immune/inflammatory response via secreted adipokines [166,173,174,175].

Two types of adipose tissue have been identified in mammals: white adipose tissue (WAT), which is the body’s main energy storage, and brown adipose tissue (BAT), which provides resistance to cold and obesity through adaptive heat production, known as nonshivering thermogenesis. BAT is a homogeneous tissue, and cells are scattered within WAT and are independent of the developmental origin; these cells are called “brown-in-white” (brite) or beige cells [176]. These brite (beige) cells are also called “inducible brown adipocytes” because they can be induced by cold and a broad spectrum of pharmacological substances [177]. In mammals, brown and beige adipocytes constitute two populations of thermogenic adipocytes with different thermogenic pathways. Thermogenic adipocytes express high levels of uncoupling protein 1 (UCP1) to dissipate energy in the form of heat by uncoupling the mitochondrial proton gradient from mitochondrial respiration. Regardless of the mechanism of thermogenesis involving UCP1, which is considered the main mechanism for BAT and crucial for systemic energy homeostasis, the existence of UCP1-independent thermogenic pathways in thermogenic adipocytes related to creatine–substrate cycling and Ca^2+^ cycling has been confirmed [178,179,180].

The development of BAT and/or brite cell induction methods may constitute a breakthrough in the treatment of obesity, as some studies have shown that BMI and body fat percentage both have significant negative correlations with BAT, whereas the resting metabolic rate has a significant positive correlation [181,182]. However, some research results limit the positive correlation of BAT volume with whole-body adiposity only to the population of young men, without confirming such a correlation in women [183].

Both WAT and BAT/brite cells express GPR120. High GPR120 expression has been detected in mature 3T3-L1 cells, a cell line often used in the study of white adipocytes [184,185,186]. Interestingly, GPR120 cannot be detected in preadipocytes, but an incubation in MDI differentiation medium containing insulin, isobutylmethylxanthine, and dexamethasone induces and increases GPR120 expression in vitro [185]. These findings and the observation that GPR120 knockdown leads to the inhibition of adipogenesis suggest the involvement of GPR120 as an adipogenic receptor in the process of adipocyte differentiation [87,187,188]. During the differentiation of 3T3-L1 adipocytes, GPR120 signaling promotes adipogenesis by increasing both peroxisome proliferator-activated receptor gamma (PPARγ) expression via Ca^2+^ influx and the phosphorylation of ERK1/2 [189].

Similarly to studies of rodents, experiments in humans revealed significantly higher expression of GPR120 in the WAT of obese individuals than in that of lean individuals [187]. This result may suggest that high lipid intake in rodents and humans induces increased expression of this receptor. In turn, increased fat oxidation in BAT observed after exposure to cold is correlated with upregulated GPR120 expression [190]. Moreover, the activation of GPR120 in BAT can promote the browning of white fat in mice [189]. The activation of the lipid sensor GPR120 is accompanied by increased secretion of fibroblast growth factor-21 (FGF-21) by BAT and brite cells, the concentration of which increases in the blood. Consistently, knockout of the FGF-21 gene causes a severe impairment of WAT browning [191].

Notably, GPR120 signaling in WAT is associated with anti-inflammatory and insulin-sensitizing effects [93]. In particular, the anti-inflammatory effects of Ω-3 FAs in WAT mediated by GPR120 are potentially therapeutically important in obesity, where low-grade chronic inflammation is a measure of adipocyte dysfunction [93,192]. This typical inflammatory state in obese individuals may most likely be a consequence of impaired signaling via GPR120 in BAT/brite cells, leading to a reduction in the BAT content as a result of the conversion of brown adipocytes to white-like unilocular cells [190,193]. Such BAT whitening can be induced by many factors, including a high ambient temperature, leptin receptor deficiency, impaired β-adrenergic signaling, and lipase deficiency, each of which is capable of inducing macrophage infiltration, brown adipocyte death, and crown-like structure (CLS) formation—a histologic marker of local inflammation with damaged or necrotic adipocytes surrounded by macrophages. A gene expression analysis revealed that BAT whitening in triglyceride lipase-deficient mice was associated with a strong inflammatory response and activation of the NLRP3 inflammasome, which consequently led to increased susceptibility of enlarged brown adipocytes to death [193]. A continuation of the above studies is necessary because the results of other experiments using GPR120-knockout mice may indicate that GPR120 signaling is not required or is not solely responsible for the anti-inflammatory and insulin-sensitizing effects mediated by Ω-3 FAs [194,195].

In summary, in adipose tissue, GPR120 plays key roles in adipogenesis, energy metabolism, and inflammation [185].

#### 3.2.2. GPR120 and Gastrointestinal Hormones

GPR120 is highly expressed in human and murine intestines and is associated with the fatty acid-induced secretion of cholecystokinin and the incretin glucagon-like peptide (GLP)-1 [196,197]. This increased expression of GPR120 in the intestinal epithelium suggests that the secretion of gastrointestinal hormones (or gut hormones) may be regulated by this receptor [97]. Disturbances in this regulation in relation to cholecystokinin (CCK) and glucagon-like peptide-1 (GLP-1), hormones that regulate appetite and food intake, promote weight loss, and improve hyperglycemia, may be important in the context of the pathomechanism of obesity [198,199,200,201,202,203]. Moreover, oil-induced secretion of another major incretin, gastric inhibitory polypeptide (GIP; also known as glucose-dependent insulinotropic polypeptide), from K cells located in the upper small intestine, is mediated by GPR120 through CCK [204,205,206,207]. As an anabolic hormone, GIP plays primary roles in lipid metabolism and the development of obesity by stimulating lipogenesis and blocking lipolysis. Chronic consumption of a high-fat diet leads to functional hyperplasia of K cells, accompanied by an increase in GIP concentrations. The results of recent studies indicate that the inhibition of the GPR120 signaling pathway in the intestine ameliorates IR and nonalcoholic steatohepatitis in mice maintained on a high-fat diet. This effect is accompanied by a reduction in GIP and CCK secretion [208,209]. However, some controversy exists, as reduced expression of GPR120/FFAR4 has been documented in cases of extreme obesity and IR in children. Researchers have speculated that this decrease in GPR120 expression, which is associated with extreme obesity and parameters indicative of obesity comorbidities in children, may be due in part to the presence of the C allele of the rs11187533 GPR120 single-nucleotide polymorphism (SNP) [210]. GPR120 gene variants associated with obesity have been identified in humans and dogs (p.Arg270His and p.Pro199Thr, respectively) [165,187].

#### 3.2.3. GPR120 and the Endocrine Function of the Pancreas

In addition to incretin hormones and glycemia regulation in the gut via the stimulation of insulin secretion after nutrient intake, GPR120 signaling affects the pancreas directly. Significant GPR120 expression was detected in the endo pancreas. Inhibitory GPR120 signaling in pancreatic islet δ cells contributes to both insulin and glucagon secretion in part by mitigating somatostatin (SRIF—somatotropin release inhibiting factor, also known as GHIH—growth hormone-inhibiting hormone) release, a powerful paracrine inhibitor of both insulin and glucagon release from islet β-cells and α-cells, respectively [211,212,213]. Consequently, GPR120 activation promotes glucose-stimulated insulin secretion (GSIS), enhances arginine-induced glucagon secretion, and inhibits glucose-stimulated somatostatin secretion (GSSS) [109,212,214].

Given that δ cells are electrically excitable, glucose can stimulate action potential firing and SRIF secretion through both metabolic and nonmetabolic effects [213,215]. In addition, membrane potential-independent pathways of glucose-induced SRIF secretion, involving cAMP-dependent stimulation of calcium-induced calcium release (CICR) and exocytosis of SRIF, are of greater quantitative importance [216].

The stimulation of insulin and glucagon secretion following GPR120 activation in δ cells constitutes an opposing mechanism to the SRIF secretion-stimulating effect of urocortin 3 (UCN3). UCN3 produced and released by pancreatic β cells activates the corticotropin-releasing factor type-2 (CRF2) receptor (CRF2R) and downstream pathways mediated by different G protein or arrestin subtypes in δ cells, resulting in SRIF gene upregulation along with the subsequent inhibition of both insulin and glucagon release, which is a part of an important feedback circuit for glucose homeostasis [217,218].

Further studies have shown that although GPR120 is preferentially expressed in pancreatic δ cells, its expression is not selective within the pancreatic islets of Langerhans, as it also affects β and α cells [164,212]. GPR120 activation in β cells reduces their dysfunction and susceptibility to apoptosis, which has a significant effect on islet hormone secretion [164,219]. For example, GPR120 signaling protects against lipotoxicity-induced pancreatic β-cell dysfunction through the regulation of pancreatic duodenal homeobox-1 (PDX1) expression and the inhibition of islet inflammation [164]. Studies examining the protective effects of polyunsaturated fatty acids (PUFAs) on β-cells have shown that docosahexaenoic acid attenuates palmitate-induced apoptosis by increasing autophagy through the GPR120/mTOR axis [220].

In obese individuals both with and without diabetes, the insulinotropic effects of GPR120 agonists are altered. Under these conditions, impaired GPR120 expression in β-cells was observed, accompanied by increased peroxisome proliferator-activated receptor γ (PPARγ) expression [221]. Promising results from the latest research on obesity and IR indicate the therapeutic potential of GPR120 activation, especially in combination with incretin (GLP-1), in enhancing the activity of dipeptidyl peptidase-IV (DPP-IV) inhibitors, which effectively stimulate the proliferation of β-cells [222,223,224,225].

## 4. Targeting GPR120 Signaling as a Promising Therapeutic Approach in Obesity: The Need for New Ligands

The use of GPR120 agonists may be beneficial in the treatment of metabolic diseases, especially those involving IR and impaired glucose tolerance, as well as chronic low-grade inflammation, which involves mainly adipose tissue in obese individuals [99,226,227,228]. Moreover, GPR120 agonists/antagonists can be used as specific anticancer therapies [229,230]. The above actions cannot be achieved selectively after the stimulation/inhibition of GPR120, but the dominant effect observed may be a result of the expression of GPR120 in a particular tissue and the lack/severity of pathological changes (e.g., intensity of the inflammatory response or progression of cancer) [32,99,231]. However, for most GPR120 agonists, separating these effects would be inappropriate, as overweight and obesity are usually accompanied by IR, T2D, and an increased inflammatory response [99,232].

The in vivo application of the anti-inflammatory and insulin-sensitizing effects associated with GPR120 stimulation, which are promising tools for the prevention and treatment of obesity, is very difficult [233]. This difficulty is caused not only by interspecies differences between the sequences encoded by the GPR120 gene in humans and laboratory animals (primarily rodents) but also by the existence of two splice variants in humans: short GPR120 (GPR120S) and long GPR120 (GPR120L) [73,75,77,79]. First, natural ligands detected by GPR120 in the form of long-chain FFAs, both saturated (14–18 carbon atoms) and unsaturated (6–22 carbon atoms), can be easily and quickly transformed (including being broken down) in vivo into other biologically active compounds [234,235]. This conversion may lead to a state in which the resulting compounds do not interact with GCPRs for LCFAs (GPR40/FFAR1 and GPR120/FFAR4) but exert biological effects through receptors for SCFAs (GPR41 and GPR43) and MCFAs (GPR84) [68,69,70,71]. Controlling the LCFA conversion process in vivo is a major challenge, which makes the results of studies on the effects of fatty acids difficult to interpret unambiguously. Moreover, as already mentioned, GPR120 expression depends on the cell type and the dietary contents of EPA, DHA, and ALA [82,236]. Therefore, the identification/isolation of new selective ligands existing in nature and, above all, the development of synthetic GPR120 ligands become crucial for a detailed and selective examination of the signaling pathways of this receptor, especially in the context of therapeutic possibilities [72,233,237,238].

With respect to the selectivity of the actions of such synthetic ligands, although GPR120 and GPR40 are physiologically activated by the same class of fatty acids (LCFAs), their structural relatedness is distant. For example, the amino acid residues involved in endogenous ligand recognition by GPR120 are similar to sphingosine-1-phosphate receptor-1 (S1P1), which is a target of the lipid signaling molecule sphingosine-1-phosphate (S1P), whereas the basis of recognition of fatty acids by GPR40 is similar to that of the SCFA receptors GPR41 and GPR43 [71].

Under in vivo conditions, researchers have not completely excluded the possibility that the anti-inflammatory effect of EPA or the increase in body weight loss in DHA-treated mice results from the activation of other molecular targets, including the transcription factors peroxisome proliferator-activated receptor alpha (PPARα) and retinoid X receptor alpha (RXRα) [72]. The conversion of fatty acids to specialized pro-resolving mediators (SPMs), such as lipoxins, resolvins, protectins, and maresins, may also be important [239]. The list of endogenous GPR120 ligands is also subject to modification. For example, a lipodomic analysis of mice overexpressing the glucose transporter 4 (GLUT4) in adipocytes revealed structures of branched fatty acid esters of hydroxy fatty acids, including palmitic acid-9-hydroxystearic acid (9-PAHSA), which act as GPR120 agonists. In adipocytes, 9-PAHSA augmented insulin-stimulated glucose uptake through GPR120 [240]. Moreover, by being involved in the antiobesity mechanism of WAT browning and increasing thermogenic activity, 9-PAHSA may become a key compound in understanding the pathogenesis of obesity. The concomitant anti-inflammatory effect of 9-PAHSA is due to the inhibition of the LPS/NF-κB pathway [241].

### 4.1. Non-LCFA GPR120 Agonists Derived from Compounds of Natural Origin

The screening of 80 natural compounds of plant origin, particularly lipids with an acidic structure, allowed the isolation of a series of grifolin derivatives, which were characterized by the ability to activate GPR120 [242]. Compounds such as grifolic acid and grifolic acid methyl ether activated ERK in cells expressing GPR120 (Flp-in GPR120). However, the potencies of these two compounds are significantly lower than that of linolenic acid (LA), which positions them as selective partial GPR120 agonists [74]. In addition, grifolic acid and grifolic acid methyl ether can act as antagonists of GPR120 by inhibiting both LA-induced ERK phosphorylation and LA-induced [Ca^2+^]i in a dose-dependent manner [242,243]. Grifolic acid, which acts via GPR120, has been shown to promote GLP-1 secretion from intestinal secretin tumor cell line 1 (STC-1) [244]. In studies analyzing the usefulness of grifolic acid in a specific therapy, assays using the macrophage-like RAW264.7 cell line revealed that grifolic acid reduced cell viability in a dose- and time-dependent manner. The apoptotic death of RAW264.7 macrophages was caused by a reduction in the mitochondrial membrane potential (MiMP) and the inhibition of ATP production via a GPR120-independent mechanism [245]. Although unrelated to the topic of this review, the ability of grifolin to induce apoptosis, cell cycle arrest, autophagy, and senescence in human cancer cell lines has attracted the attention of researchers searching for anticancer drugs [246]. An exception of uncertain significance may be apoptosis, which, under conditions of preserved homeostasis, occurs without an inflammatory reaction because the activation of caspases ensures that inflammatory pathways are disabled. However, under specific, altered conditions accompanying obesity, apoptosis and the apoptotic machinery can be rewired into a process that is inflammatory [247,248].

The common hop or hops (*Humulus lupulus* L.) became the source of KDT501, the potassium salt of the n-(isobutyl) congener of a tetrahydroiso-α-acid, also known as an isohumulone drug [249]. This PPARγ and GPR120 agonist was tested in a small group (n = 9) of obese patients, and thus far, KDT501 is the only GPR120 agonist that has entered phase II clinical trials. After considering the short duration of the study, the small study group, and the lack of selectivity of KDT501, its effectiveness in treating IR, obesity, or metabolic syndrome can be confirmed, as in all cases, a reduction in the levels of systemic inflammation markers and improved postmeal plasma triglyceride levels were observed [250]. Moreover, KDT501 post-transcriptionally induced adiponectin secretion from subcutaneous WAT and increased the expression of thermogenic and lipolytic genes in response to cold exposure [251].

Teadenol A, a polyphenol (1-benzopyran) isolated from Japanese postfermented tea (i.e., produced by microbial fermentation), acts as a novel GPR120 ligand that binds directly and induces ERK1/2 phosphorylation and intracellular Ca^2+^ store activation with cytoplasmic Ca^2+^ influx [252]. In a human embryonic kidney (HEK) cell line (293T), these effects were completely dependent on GPR120 expression and correlated with teadenol A-induced GPR120 upregulation. These effects were abolished by the use of the selective GPR120 antagonist AH-7614 (Compound **39**). In addition, teadenol A increased the secretion of GLP-1 from intestinal endocrine STC-1 cells, which indicates its indirect effects on metabolic homeostasis, as GLP-1 improves glycemic control and stimulates satiety (suppresses appetite), leading to reductions in food intake and body weight [252].

The characteristics of the substances of natural origin listed above, including the possible therapeutic use of their GPR120 agonist activities in the context of metabolic changes accompanying obesity, are summarized in Table 1.

### 4.2. GPR120 Synthetic Agonists

The use of new techniques for computer modeling of the spatial structure of ligands for GPR120 has expanded the potential group of compounds functioning as agonists of this receptor to a previously unavailable scale. For example, in the search for peptides that bind to GPR120, a virtual library was created in which 531,441 low-polarity hexapeptides were collected [224]. The screening of these more than 500,000 peptides, which involved computational work starting with the narrow filtering of hexapeptides based on their chemical similarity to known GPR120 agonists, provided valuable mechanistic insights into GPR120 activation and further extended the knowledge of GPR120 stereochemistry, which is crucial for agonistic ligand recognition. The best hits were tested through docking studies, molecular dynamics, and umbrella sampling simulations, which identified G[I,L]FGGG [the binding affinity (ΔG) of GIFGGG: −6 kCal/mol] as a promising GPR120 agonist sequence. Additionally, D-amino acids may increase the peptide–GPR120 interaction (ΔG of GdIFGGG: −13 kCal/mol) [224]. In the development of predictive models for better real-time monitoring of GPR120–ligand interactions at the nanomolar level in the context of designing new synthetic agonists, the use of fluorescence is an important aid. The use of ligands with fluorescent properties and the high viscosity and low polarity of the ligand binding area, which also show environmental sensitivity, enables the detection of small-molecule sulfonamide agonists for GPR120 [267].

The current classic approach to developing selective, potent GPR120 agonists involves the use of existing crystal structures of other GPCRs [72,225,243,268]. In this way, the key to the activity of the agonist toward GPR120 is the formation of hydrogen bonds between the carboxylic acid residue of the ligand and the guanidine of Arg99 in GPR120 [228]. However, the greatest problem was ensuring high selectivity of the synthetic ligand, especially when GPR40/FFAR1 is coexpressed.

The first synthetic GPR120 agonist was GW9508, although it was originally developed as a highly selective GPR40/FFAR1 orthosteric agonist [269]. GW9508 shows ~100-fold selectivity for GPR40 over GPR120. This selectivity corresponds to the pEC50 values for GW9508, which, in relation to GPR40 and GPR120, are 7.32 and 5.46, respectively [269]. However, GW9508 also activated GPR120 at relatively high concentrations. The weak affinity and lack of selectivity of GW9508 for GPR120 limits the use of this compound in studies of GPR120 stimulation. In the absence of other synthetic ligands, GW9508 is used as a GPR120 antagonist in cell systems in which GPR40 is not detected because of GPR40 knockout (e.g., the RAW 264.7 monocyte-/macrophage-like cell line) [93,196]. GW9508 is inactive against other GPCRs, kinases, proteases, integrins, and PPARs. Its potential therapeutic effects (e.g., anti-atherosclerotic and anti-inflammatory effects) involve promoting insulin secretion depending on the glucose concentration and opening of ATP-sensitive potassium (KATP) channels [270].

Another trend in the search for GPR120 agonists has been attempts to modify PPAR-γ agonists with N-carbamoyl-beta-D-glucopyranosylamine (NCG). NCG21, a PPAR-γ agonist derivative, shows high (albeit incomplete) selectivity for GPR120 over GPR40 [225]. The problem of very limited selectivity remains; therefore, NCG21 can be used with cell lines and tissues expressing GPR120 in the absence of GPR40 expression to study the role of GPR120 in physiological processes [243]. NCG21, the NCG compound with the lowest calculated hydrogen bonding energy, was determined to be a very strong extracellular signal-regulated kinase (ERK) activator in a cloned GPR120 system. In addition, in murine enteroendocrine STC-1 cells endogenously expressing GPR120, NCG21 potently activated ERK, intracellular calcium responses, and GLP-1 secretion. In the mouse colon, an increase in plasma GLP-1 levels was positively correlated with intracolonic NCG21 administration [243,271].

Extensive research by Shimpukade et al. [268] on the relationship between the spatial structures and biological activities of new compounds led to the identification of TUG-891, a derivative of the GPR40 agonist GW9508. Unlike GW9508, TUG-891 proved to be the first GPR120 agonist with high activity and 1000-fold selectivity over human GPR40 in assays based on the induced interactions between the receptor and β-arrestin 2 [268]. TUG-891 has high potency against both human and mouse GPR120 and is a more selective and potent agonist of human GPR120 than ALA, GW9508, or NCG21 [272]. Initially, TUG-891 was widely used in research on the physiological function of GPR120. More extensive studies, however, revealed that the selectivity between human GPR120 and GPR40 was significantly less pronounced in mouse orthologs and variable when G protein-mediated increases in Ca^2+^ influx versus receptor interactions with arrestin were measured [233]. This difference in the selectivity of TUG-891 between humans and mice is likely not due to the decreased potency of mouse GPR120 but rather to the increased potency of mouse GPR40. Therefore, in practice, TUG-891 should be treated as a strong and selective GPR agonist in studies using human cells and tissues, whereas in the equivalent rodent tissues, TUG-891 should be viewed as a dual GPR120 and GPR40 agonist [233,272]. Hence, the potential utility of TUG-891 in animal studies may be restricted. Additionally, limitations in the use of TUG-891 result from its poor metabolic stability and high lipophilicity (cLogP = 5.88) due to the susceptibility of phenylpropanoic acid to β-oxidation and the presence of a biphenyl structure in the molecule with high lipophilicity [273]. Derivative compounds have been successfully developed to eliminate susceptibility to β-oxidation and the associated loss of GPR120 agonistic activity by TUG-891 [274].

The importance of GPR120 in the regulation of glucose homeostasis, especially through its effect on insulin sensitivity, was proven in studies of highly selective GPR120 agonists in rodents: a phenylpropanoic acid (Compound **29**)-based series and a chromane propionic acid (Compound **18**)-based series. Compared with GPR40, Compound **29** derivatives showed at least 300-fold greater selectivity for both human and mouse GPR120 [275]. Additionally, Compound **18** derivatives showed significantly increased GPR120 selectivity, which exceeded GPR40 selectivity by 40–130 times across human, mouse, and rat orthologs [276].

Merck (Darmstadt, Germany) patented a highly selective, nontoxic, cell-permeable, and reversible GPR120 superagonist (EC50~0.35 µM) called Compound A (cpdA), which is available in oral form. Compound A is approximately 50-fold more effective than DHA in stimulating GPR120-related responses [277]. The affinity of cpdA for GPR40/FFAR1 and thus the degree of activation of this receptor were negligible. Compared with ω3-FA administration, treatment with cpdA in vitro and in vivo caused concentration-dependent anti-inflammatory and insulin-sensitizing effects [82]. These effects were observed in high-fat-diet-fed obese mice when cpdA was administered at a dose of 30 mg/kg [187,277]. The anti-inflammatory effect was observed in macrophages, where cpdA strongly inhibited LPS-induced TAK1, IKKβ, and JNK phosphorylation and reduced IκB degradation [274]. Using cpdA, researchers have also confirmed the beneficial immunomodulatory effects of DHA on the treatment of atopic dermatitis, which involves an increase in the Foxp3+ Treg population after GPR120 stimulation [278].

Metabolex 36 (3-(3,5-difluoro-4-((3-methyl-1-phenyl-1H-pyrazol-5-yl)methoxy)phenyl)-2-methylpropanoic acid) was developed by the company Metabolex (currently Cymabay Therapeutics) as a potent, selective agonist of human and mouse GPR120. The selectivity of Metabolex 36 toward GPR120 was confirmed by the lack of GPR40/FFAR1 activation, as determined via a calcium mobilization assay [279]. The dynamic mass redistribution (DMR) response and cAMP production were induced by Metabolex 36 in GPR120-overexpressing cells. However, in mouse islets, this compound inhibits cAMP production, and in a murine neuroendocrine cell line (STC-1), Metabolex 36 increases GLP-1 secretion. In studies of lean mice, Metabolex 36 (30 mg/kg) improves oral glucose tolerance and increases insulin secretion during the intravenous glucose tolerance test (IVGTT). This GPR120 agonist inhibits glucose-induced somatostatin secretion in a dose-dependent manner when murine pancreatic islets are incubated under either maximal stimulating conditions (16.6 mmol/L) or with intermediate glucose concentrations (8 mmol/L). Maximal inhibition was achieved with 30 μmol/L of the ligand. Under both basal conditions and in the presence of a stimulating glucose concentration (16.6 mmol/L), Metabolex 36 has no effect on insulin secretion [211].

Developed later than Metabolex 36, the compound AZ13581837 was a selective human and mouse GPR120 agonist and was able to induce an approximately 100-fold greater DMR response in CHO-hGPR120/mGPR120 cells (EC50 values of 5.2 nM and 4.3 nM, respectively) than Metabolex 36 [279]. AZ13581837 selectively induces a calcium response in CHO-hGPR120 cells with an EC50 of 120 nM and induces GPR120-dependent calcium mobilization and cAMP production in GPR120-overexpressing cells. Like Metabolex 36, AZ13581837 (10 µM) inhibits cAMP production in mouse islets and increases GLP-1 secretion from STC-1 cells. When administered at a dose of 18 mg/kg, AZ13581837 improves oral glucose tolerance and increases insulin secretion in lean mice, according to IVGTTs [212,280].

Another synthetic agonist, GPR120 agonist III (3-(4-((4-fluoro-4′-methyl-(1,1′-biphenyl)-2-yl)methoxy)-phenyl)propanoic acid), is fully selective for GPR120 (pEC50 = 7.62) and has negligible activity toward GPR40 [277]. Under the influence of GPR120 agonist III, a concentration-dependent increase in IP3 production was detected in human and mouse cells expressing GPR120, as well as a concentration-dependent response in which β-arrestin-2 was recruited (EC50s~0.35 μM). The significant anti-inflammatory effect of GPR120 agonist III is manifested by the inhibition of the LPS-induced phosphorylation of TAK1, IKKβ, and JNK and the blockade of IκB degradation [281]. To date, the increase in insulin sensitivity induced by GPR120 III agonist at increased glucose infusion rates, the increased rate of insulin-stimulated glucose disposal, and the increased ability of insulin to inhibit hepatic gluconeogenesis have been documented only in WT mice. The use of GPR120 agonist III may have a beneficial effect on hepatic lipid metabolism because it leads to decreased hepatic steatosis and decreased liver triglycerides and DAG levels, along with a reduced saturated FFA content [282].

In light of the research that led to the development of GSK137647A, GSK137657A, and TUG-1197 (also known as Compound **34**), the presence of a carboxylate that directly interacts with Arg99 of GPR120 is not a sine qua non condition for agonist properties. Unlike most synthetic agonists with a structural similarity to free fatty acids, GSK137647A, GSK137657A, and TUG-1197 are potent nonacidic sulfonamide-containing GPR120 agonists [233,267,283,284]. TUG-1197 has no detectable activity toward GPR40, while GSK137647A shows over 50 times greater selectivity for GPR120 than GPR40, and this selectivity is maintained across species [233]. The finding that the selectivity of GSK137647A toward GPR120 is related to its phenyl ring-conjugated structure was the starting point for the development of new types of agonists based on the core structure of GSK137647A [267]. For example, the sulfonamide agonist GSK137657A has high selectivity and activity in promoting insulin secretion [267]. The lack of the acidic nature of the above compounds did not prevent easy binding within the same orthosteric binding pocket as the carboxylate-containing agonists that resemble synthetic fatty acids [74,233]. The effects of GSK137647A and TUG-1197, among others, on the hypothalamus of obese individuals were tested to assess the impact of GPR120 stimulation on energy homeostasis and GPR120-induced anti-inflammatory effects [285,286]. Even if the poor pharmacokinetic and pharmacodynamic properties of GSK137647A and TUG-1197 are considered, the preliminary results are promising in terms of the possibility of identifying new pharmacological targets for metabolic diseases. However, despite extensive in vitro and in vivo animal studies, none of these compounds, and none of the currently known synthetic selective GPR120 agonists, have entered regular clinical trials with a large number of patients [274].

The characteristics of the selected synthetic agonists listed in this section, including the potential therapeutic use of their GPR120 activities in the context of the metabolic changes accompanying obesity, are summarized in Table 2.

## 5. In Summary: Challenges in the Therapeutic Use of GPR120 Agonists and Future Strategies

Table 3 summarizes the fundamental difficulties in developing and implementing GPR120 agonist-based treatments for metabolic disorders in obesity and indicates likely future strategies in this area. The limitations of naturally occurring and synthetic agonists are often similar, while future prospects differ significantly in favor of synthetic ligands.

## 6. Concluding Remarks

Recent studies have shown that FFAs not only are sources of energy in the body as nutritional components but also serve as ligands for a group of orphan GPCRs called free fatty acid receptors (FFARs), which activate numerous signaling pathways. Among FFARs, special attention is given to those activated by long-chain saturated and unsaturated FFAs, i.e., GPR40/FFAR1 and GPR120/FFAR4, especially the latter.

Undoubtedly, signaling through GPR120/FFAR4 significantly affects the body’s metabolism and plays a key role in maintaining energy homeostasis in both humans and rodents. Glucoregulatory and insulin-sensitizing/insulin-releasing activities have come to the forefront. They become particularly important in obesity, where metabolic disorders are characterized by IR and glucose intolerance, with an increased incidence of diabetes. Moreover, the high expression of GPR120 in adipocytes, pancreatic islet cells, and enteroendocrine L and K cells of the gastrointestinal tract enables the definition of therapeutic goals that can be achieved using GPR120 agonists. Examples include stimulating thermogenesis in BAT and/or the browning of WAT, promoting direct beneficial effects on pancreatic β-cell health (e.g., anti-inflammatory and immunomodulatory effects), and modulating the secretion of hormones from the gastrointestinal tract and pancreas, including GLP-1/GIP secretory pathways. The expected reduction in chronic low-grade inflammation in adipose tissue associated with GPR120 stimulation should additionally work toward restoring normal metabolism.

Owing to the detailed knowledge of the structure, particularly the 3D conformation of GPR120, the relatively small pool of naturally occurring GPR120 agonists is constantly expanding through the almost unlimited possibilities of developing synthetic ligands. A properly designed structure of such a ligand is likely to direct its action in a more specific way, e.g., with a dominant anti-inflammatory effect, stronger stimulation of thermogenesis, or the promotion of the secretion of incretin hormones and insulin. However, the biological properties of these new GPR120 agonists and/or antagonists must be comprehensively tested in terms of potency, selectivity, or dominant effects under specific clinical conditions as well as in terms of side effects and safety in humans. Current knowledge of their effects on humans is still insufficient, and large-scale clinical trials using synthetic GPR120 agonists have not been conducted. Interspecies differences mean that the results of studies using rodents should be interpreted with caution, but interindividual differences in humans may also be important.

The repeatability of metabolic effects on regulating glycemia and changes in adipose tissue may be influenced by the lack of selectivity of the action of a given agonist. Notably, the lack of selectivity for GPR120 alone does not determine the therapeutic usefulness of a given agonist because, for example, the metabolic effects of simultaneous GPR40 stimulation are largely similar to those observed after GPR120 stimulation and are generally beneficial for obese individuals. The key to the effectiveness of these FFAR agonists is their ability to activate adipose tissue by inducing changes in mitochondrial dynamics, resulting in increased O_2_ consumption, increased nutrient uptake, and a reduction in the fat mass. Just reducing fat mass should lead to a decrease in the intensity of the inflammatory reaction and improved insulin sensitivity. However, at the current stage, the development of drugs based on GPR120 agonists for the treatment of patients with obesity-related disorders requires conducting many large-scale clinical trials and solving the above-mentioned problems.

## Figures and Tables

**Figure 1 ijms-26-02501-f001:**
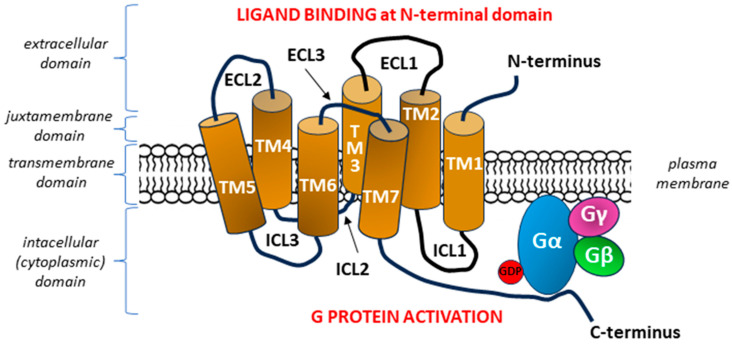
General diagram of the structure of a G protein-coupled receptor (GPCR). The seven transmembrane heptahelical structures (TM1–TM7) are accompanied by three extracellular loops (ECL1–ECL3) on the N-terminal side of the molecule and three intracellular loops (ICL1–ICL3) on the side of C-terminal tail [29,30]. After binding the signal molecule (ligand) at the N-terminal domain, conformation changes occur in the transmembrane GPCR molecule, which enable ICLs to interact with the G protein within the C-terminus located in the intracellular domain, with subsequent activation of the G protein [34]. Each G protein is composed of three subunits α, β, and γ with a nucleotide-binding pocket located in the α subunit. In the inactive heterotrimeric state, guanosine diphosphate (GDP) is bound to the Gα subunit. The formation of the GPCR—G protein complex after GPCR stimulation begins with the release of GDP from its binding site on the G alpha subunit, which is equivalent to the activation of the G protein, as it allows the binding of guanosine triphosphate (GTP) and inducing the dissociation of the α subunit of the G protein from the βγ subunits [35,36].

**Figure 2 ijms-26-02501-f002:**
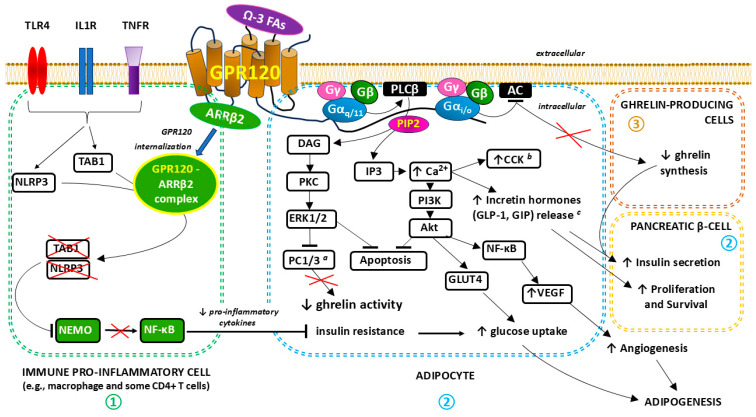
The three main signal transduction pathways associated with GPR120/FFAR4 stimulation: 
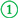
—**ANTI-INFLAMMATORY EFFECTS via beta-arrestin 2 (ARRβ2) signaling in immune pro-inflammatory cells.** After receptor stimulation, the adapter protein ARRβ2 is recruited, which, after forming a complex with GPR120, leads to the desensitization of GPR120 as well as other receptors [e.g., Toll-like receptor 4 (TLR4), interleukin 1 receptor (IL1R), and tumor necrosis factor receptor (TNFR)] to pro-inflammatory stimuli. The GPR120- ARRβ2 complex internalizes and interacts with TGF-β-activated kinase 1 (MAP3K7)-binding protein 1 (TAB1) and NLRP3 [nucleotide-binding oligomerization domain (NOD), leucine-rich repeat (LRR)-containing protein (NLR) family member 3] inflammasome, which leads to the inhibition of the subsequent inflammatory cascade. Nuclear factor kappa-light-chain-enhancer of activated B cells (NF-кB) essential modulator (NEMO) ubiquitination, which is crucial for the activation of the canonical NF-κB signaling pathway, does not occur. A decrease in the concentration of pro-inflammatory cytokines indirectly modulates adipose tissue cell functions by reducing insulin resistance and, consequently, stimulating glucose uptake [82,94,95,96,97,98]. 
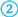
—**METABOLIC EFFECTS via classical Gα q/11 signaling**. In this pathway, omega-3-fatty acids (Ω-3 FAs), acting as a ligand and agonist of GPR120, activate the Gαq/11-linked phospholipase C-beta (PLCβ). After PLCβ activation, phosphatidylinositol 4,5-bisphosphate (PIP2) hydrolyzes into diacyl glycerol (DAG) and inositol 1,4,5-trisphosphate (IP3), which serve as second messengers. IP3 binds to receptors, the conformational changes of which, after activation, lead to the mobilization of calcium ions (Ca^2+^) from intracellular stores. The following cytosolic Ca^2+^ influx stimulates the release of both cholecystokinin (CCK) and incretin hormones, including glucagon-like peptide-1 (GLP-1) and gastric inhibitory polypeptide (GIP, also known as glucose-dependent insulinotropic polypeptide). Incretin hormones support the survival and proliferation of pancreatic β-cells, which translates into increased insulin secretion. Moreover, the increase in Ca^2+^ concentration activates the phosphoinositide 3-kinase (PI3K) pathway including Akt, which stimulates downstream anabolic signaling, favoring cell growth (adipogenesis), proliferation, (angiogenesis), and survival. Upregulated glucose transporter type 4 (GLUT4) activity promotes increased glucose uptake, whereas augmented vascular endothelial growth factor (VEGF) expression induced by NF-кB enhances angiogenesis. Increased DAG synthesis activates protein kinase C (PKC), which results in the further activation of various intracellular signal transduction systems, including extracellular signal-regulated kinases ERK1 and ERK2 (ERK1/2). ERK1/2 has anti-apoptotic effects promoting cell growth and survival and indirectly reduces ghrelin activity by inhibiting proprotein convertase 1/3 (PC1/3), an enzyme necessary for the cleavage of pro-ghrelin into mature ghrelin in the Golgi apparatus [91,92,93,96,99,100,101,102,103,104]. 
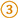
—**METABOLIC EFFECTS via Gα i/o signaling**. Gαi/o affects related signaling pathways through the inhibition of adenylate cyclase (AC) activity, influencing intracellular cyclic adenosine 3,5-monophosphate (cAMP) levels in ghrelin-producing cells with a subsequent decrease in ghrelin synthesis. In this mechanism, the inhibitory effect of ghrelin on insulin secretion in pancreatic islet β-cells is reduced [99,105]. ^a^ PC1/3 is synthesized and expressed in a variety of tissues such as the neuroendocrine cells in the arcuate and paraventricular nuclei of the hypothalamus, the enteroendocrine cells (gut), and in β cells of the pancreas. ^b^ CCK is produced in the intestinal epithelial endocrine I-cells of the small intestine (mainly in the duodenum) and various neurons in the gastrointestinal tract and central nervous system (CNS). ^c^ GLP-1 and GIP are produced in the intestinal epithelial endocrine L cells and K cells, respectively. Since 2003, when Fredriksson et al. discovered the GPR120/FFAR4 protein [43,106], interest in this receptor has consistently increased as further reports of the anti-inflammatory and metabolic effects of its agonists have appeared [99,107,108,109,110,111,112].

**Figure 3 ijms-26-02501-f003:**
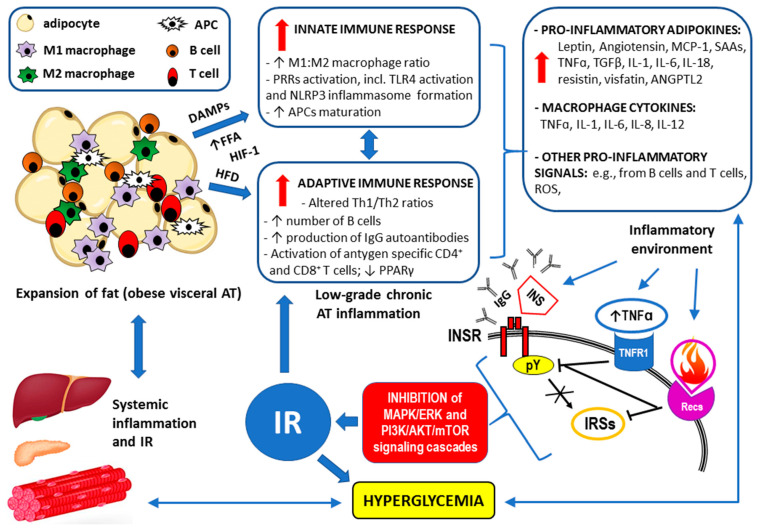
Summary of the mechanisms by which visceral obesity is a triggering factor for both systemic inflammation and insulin resistance (IR). Adopted from [28]. Strongly enlarged adipocytes in the obese visceral adipose tissue (AT) fail to maintain longer metabolic homeostasis because the lipid overload leads to endoplasmic reticulum stress, increased expression of the inflammation regulator NF-kB, and the production of inflammation-inducing signals [135]. This chronic metabolism-induced inflammation or meta-inflammation in obese AT activates the resident immune cells, including macrophages, B cells, T cells, and antigen-presenting cells (APCs) [145]. A high-fat diet (HFD), an increased flux of free fatty acids (FFAs) accumulating damage-associated molecular patterns (DAMPs) from necrotic AT, and hypoxia-induced factor 1 (HIF-1) trigger both innate and adaptive immune responses [135]. The resulting low-grade chronic AT inflammation manifests as significantly elevated levels of the pro-inflammatory adipokines and cytokines as well as through the overproduction of reactive oxygen species (ROS) [117,126,130,146,147,148,149,150]. Such an inflammatory environment interferes with insulin signaling via the insulin receptor (INSR) through antibodies/autoantibodies against both insulin (INS) and INSR [151,152]; tumor necrosis factor receptor (TNFR1) through the inhibition of tyrosine autophosphorylation (pY) within the INSR and/or serine/threonine phosphorylation (pS) of the insulin receptor substrates (IRSs) [152,153,154,155]; and receptors for other pro-inflammatory cytokines (Recs) through abnormal, similar to that observed for TNFR1 signaling, phosphorylation of the INSR and IRSs [153,154,156]. Disrupted downstream signaling via the mitogen-activated protein kinase/extracellular signal-regulated kinase (MAPK/ERK) and phosphoinositide 3-kinase/protein kinase B/mammalian target of rapamycin (PI3K/AKT/mTOR) pathways eventually lead to IR. The resulting hyperglycemia may increase the inflammatory response by itself and lead to systemic inflammation [157,158,159]. The remaining abbreviations: ANGPTL2—angiopoietin-like 2 protein; IL-1, IL-6, IL-8, IL-12, IL-18—the respective interleukins; MCP-1—monocyte chemoattractant protein-1 (also known as CCL2); NLRP3 inflammasome—leucine-rich repeat (LRR)-containing proteins (NLR) family member 3 inflammasome; SAAs—sulfur amino acids; TGFβ—transforming growth factor β.

**Table 1 ijms-26-02501-t001:** Examples of naturally occurring GPR120 agonists with potentially beneficial effects in obesity-related metabolic disorders.

GPR120/FFAR4 Non-LCFA Agonist from Natural Sources	The Structure of the Molecule and Chemical Nomenclature	Source	Agonist’s Potency (EC50) *	Selectivity	Documented Metabolic Effects in Obesity-Related Disorders
**Grifolin derivatives:** **- Grifolic acid** and **- Grifolic acid methyl ether**	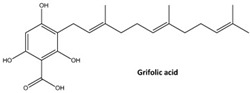 2,4-dihydroxy-6-methyl-3-[(2E,6E)-3,7,11-trimethyldodeca-2,6,10-trienyl] benzoic acid 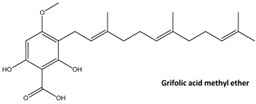 2-hydroxy-4-methoxy-6-methyl-3-[(2E,6E)-3,7,11-trimethyldodeca-2,6,10-trienyl] benzoic acid	First extracted from the mushroom *Albatrellus confluens*, its content was also confirmed in *Albatrellus dispansus*, *Albatrellus ovinus*, *Peperomia galioides*, a species of plant in the family *Piperaceae*, and other organisms with data available (i.e., Rhododendron *dauricum*), including tomato (*Solanum lycopersicum* L.) [246,253,254]	>30 µM [242,243,255] >30 µM [242,253,255]	Selective partial agonist Selective partial agonist	- Antioxidative activity properties more potent than both α-tocopherol and butylated hydroxyanisole (BHA) [256,257]. - As partial agonists with high selectivity, grifolin derivatives mediate the anti-inflammatory effects of n-3 polyunsaturated fatty acids (PUFAs); however, they are less efficacious in activating GPR120 compared to ALA. Additionally, although acting in a GPR120-independent manner, grifolic acid may reduce inflammation by stimulating apoptosis (a non-inflammatory event) of cells involved in the inflammatory reaction, including macrophages [242,245,248,253,258,259]. - By acting on GPR120 expressed on neuroendocrine intestinal L cells and K cells, grifolin derivatives stimulate GLP-1/GIP secretory pathways involved in mediating enhanced insulin secretion and improved glucose tolerance [201,205,206]. - Contrary to in vitro conditions, grifolin derivatives after oral administration in WT mice increase ghrelin secretion and its concentration in plasma. Next, ghrelin secreted from GPR120 (+) cells can sense LCFAs and MCFAs directly. Thus, the GPR120-related lipid-sensing cascade of the ghrelin cell, despite reduction in inflammation, may be responsible for the increases in adipogenesis, gluconeogenesis, triglyceride, and FA synthesis as well as increased appetite and food intake [260,261].
**KDT501 (isohumulone)**	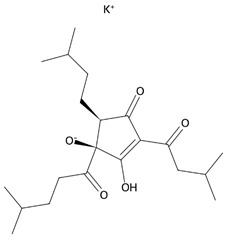 potassium;(1R,5S)-2-hydroxy-3-(3-methylbutanoyl)-5-(3-methylbutyl)-1-(4-methylpentanoyl)-4-oxocyclopent-2-en-1-olate	Substituted 1,3-cyclopentadione chemically derived from hops (*Humulus lupulus* L.) extracts [249]	30.3 µM (mice) [249]	Non-selective, also partial PPARγ agonist [249]	- Oral administration of KDT501 in diet-induced obese (DIO) mice and in a standard model for metabolic syndrome and T2D, the Zucker Diabetic Fatty (ZDF) rats, improved glucose metabolism and reduced plasma hemoglobin A1C (HbA1c) content [72,249]. - Dose-dependent reduction in weight gain and total cholesterol was demonstrated in ZDF rats receiving KDT501 [249]. - When high doses of KDT501 were used in rodent models of diabetes, weight loss was observed that was equivalent to a significant reduction in fat mass [249]. - No effect of KDT501 on lipogenesis was observed during DHA administration [72,249]. - In a small study (n = 9) of patients with obesity and prediabetes, treatment with KDT501 to a maximum dose of 1000 mg every 12 h for a total of 28 days demonstrated a reduction in systemic inflammatory markers and improvement in postmeal plasma triglyceride levels [250]. - KDT501 induced post-transcriptional secretion of adiponectin and increased the gene expression of thermogenic and lipolytic genes in response to cold stimulation [251]. - The assessment of the effects should take into account the lack of selectivity towards GPR120 and stimulation of PPAR-γ [249].
**Teadenol A**	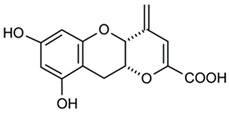 (4aR,10aR)-7,9-dihydroxy-4-methylidene-10,10a-dihydro-4aH-pyrano[3,2-b]chromene-2-carboxylic acid	Polyphenol isolated from both Japanese and Chinese postfermented teas (*Camellia sinensis* L.) [252]	N.A.D. ^#^	Selective agonist [252]	- The beneficial effects of teadenol A observed in metabolic syndrome were completely dependent on GPR120 expression on target cells [252,262]. - Under the influence of teadenol A, the secretion of GLP-1 from intestinal endocrine STC-1 cells increases—consequently, GLP-1 suppresses appetite and increases insulin secretion, exhibiting anti-diabetic effects [252,263]. - Teadenol A-containing teas are expected to improve fasting blood glucose levels and IR because teadenol A promotes the secretion of adiponectin from adipocytes [252,264]. - Microbial fermentation containing teadenol A improves triglyceride levels in prediabetic subjects, which is most likely due to the promotion of adiponectin secretion, a well-known adipokine that lowers triglycerides and LDL cholesterol levels [264,265,266].

* based on Ca^2+^ mobilization (EC50); values for human FFAR4/GPR120 unless noted; ^#^ N.A.D.—denotes no data or ambiguous data.

**Table 2 ijms-26-02501-t002:** General characteristics of selected synthetic GPR120 agonists showing potentially beneficial effects in obesity-related metabolic disorders.

Synthetic GPR120/FFAR4 Agonist	The Structure of the Molecule and Chemical Nomenclature	Agonist’s Potency (EC50) *	Selectivity	Documented Metabolic Effects in Obesity-Related Disorders
**GW9508**	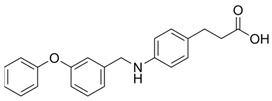 4-{[(3-phenoxyphenyl)methyl]amino} benzenepropanoic acid	2.2 µM [91]	Non-selective: ~100-fold selectivity for GPR40 over GPR120	- Decrease in ghrelin secretion [287]. - Increase in the thermogenic activity of BAT and WAT [191]. - Increase in glucose-induced insulin secretion with a subsequent reduction in plasma glucose [214,288]. - Increased insulin sensitivity [270]. - Induction of the release of fibroblast growth factor 21 (FGF-21) by brown and beige adipocytes to maintain glucose homeostasis with a subsequent increase in blood FGF-21 levels to improve metabolic adaptation to fasting [191]. - Inhibition of inflammatory responses in obesity, including improvement of metabolic syndrome-exacerbated periodontitis in mice [270,289,290].
**NCG21**	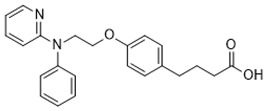 4-{4-[2-(phenyl-pyridin-2-yl-amino)-ethoxy]-phenyl}-butyric acid	N.A.D.^#^ [225,233,243,271]	PPAR-γ agonist derivative; high (albeit incomplete) selectivity for GPR120 over GPR40	- Increase in GLP-1 secretion [243]. - Intracolonic administration of NCG21 in mice was positively correlated with increased plasma GLP-1 levels [243,271].
**TUG-891**	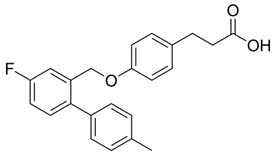 ortho-biphenyl ligand 4-{[4-fluoro-4′-methyl(1,1′-biphenyl)-2-yl]methoxy}-benzenepropanoic acid	73.74 μM, (bovine neutrophils) [291]	GPR40 non-selective agonist GW9508 derivative; a potent and selective agonist of the GPR120 with 1000-fold selectivity over human GPR40	- Amelioration of meta-inflammation and IR by increased visceral WAT p-Akt/Akt responses to insulin [292]. - Prevention of increased food intake and weight gain in mice undergoing chronic sleep fragmentation [292]. - Increase in fat oxidation and reduction in fat mass by stimulation of mitochondrial respiration in BAT [190,293]. - Increase in GLP-1 secretion and decrease in circulating LDL [294]. - Promoting the differentiation of 3T3-L1 adipocytes, i.e., adipogenesis from preadipocytes to mature adipocytes [189]. - Protective effect against lipotoxicity-induced pancreatic β-cell dysfunction, via the mediation of the pancreatic and duodenal homeobox 1 (PDX1) expression and inhibition of islet inflammation [170]. - Activation of thermogenesis by activation of UCP1, which constitutes one of the primary mechanisms by which BAT increases energy expenditure [190,293].
**Compound 29 (phenylpropanoic acid)**	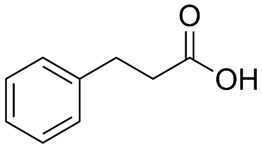 3-Phenylpropanoic acid	42 nM [295]	Highly selective, medium potency	- Increased insulin sensitivity [275,276].
**Compound 18 (chromane propionic acid derivative)**	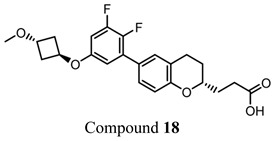	35 nM (mice) [275]	Highly selective and potent	- Increased insulin sensitivity [275].
**Compound A (Merck)**	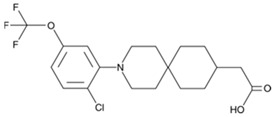 3-[2-chloro-5-(trifluoromethoxy)phenyl]-3-azaspiro[5.5]undecane-9-acetic acid	~0.35 µM (mice) [277]	Highly selective, reversible superagonist	- Anti-inflammatory effects in macrophages in vitro [277]. - Improvement in glucose tolerance and decrease in hyperinsulinemia [277]. - When administered orally in obese mice on a high-fat diet, concentration-dependent anti-inflammatory and insulin-sensitizing effects of comparable potency to ω3-FA administration were observed [82]. - Reduction in fatty liver disease [277].
**Metabolex 36**	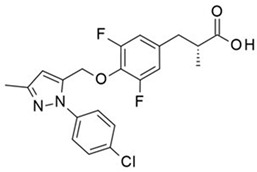 3-(3,5-difluoro-4-((3-methyl-1-phenyl-1H-pyrazol-5-yl)methoxy)phenyl)-2-methylpropanoic acid	570 nM [211]	Selective and potent	- Improved oral glucose tolerance and increased insulin secretion in lean mice during the intravenous glucose tolerance test (IVGTT) [211,296]. - Suppression of glucose-induced somatostatin secretion from pancreatic δ cells [211]. - Inhibition of cAMP production in mouse islet cells and increased production and secretion of GLP-1 from intestinal neuroendocrine STC-1 cells [279,296].
**AZ13581837**	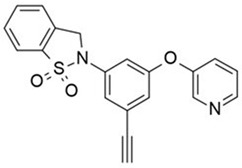 2-(3-Ethynyl-5-(3-pyridyloxy)phenyl)-3H-1,2-benzothiazole 1,1-dioxide	120 nM [279]	Selective and potent	- Inhibition of cAMP production in mouse islet cells and increased production and secretion of GLP-1 from intestinal neuroendocrine STC-1 cells [212,279] and pancreatic δ cells [211]. - Improved oral glucose tolerance and increased insulin secretion in IVGTT in lean mice [212,280].
**GPR120 agonist III**	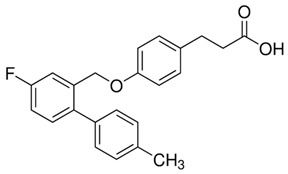 3-(4-((4-Fluoro-4′-methyl-(1,1′-biphenyl)-2-yl)methoxy)-phenyl)propanoic acid	44 nM [268]	Fully selective for GPR120 and potent	- Improvement in insulin sensitivity, overcoming IR [277]. - Stimulating the ability of insulin to inhibit gluconeogenesis in the liver [277,282]. - Significant anti-inflammatory activity manifested by the inhibition of LPS-induced phosphorylation of TAK1, IKKβ, and JNK, and blocked IκB degradation [281]. - Beneficial effect on hepatic lipid metabolism by lowering the level of hepatic triglycerides and DAGs, reducing the content of saturated FFAs and reducing liver steatosis [282].
**GSK137647A**	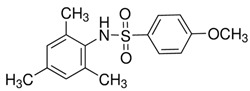 4-Methoxy-N-(2,4,6-trimethylphenyl)-benzenesulfonamide, N-Mesityl-4-methoxybenzenesulfonamide	501 nM (mice) [297]	Selective and potent	- Participation in glycemic homeostasis by inducing insulin secretion and inhibiting epithelial ion transport [298]. - Protection of pancreatic β cell dysfunction by inhibiting islet inflammation [164]. - Anti-inflammatory response accompanied by a reduction in NO production by macrophages [298]. - Suppression of adipogenic differentiation of mesenchymal stem cells [297]. - Anti-inflammatory effect in adipose tissue related to the inhibition of the NF-κB pathway and, consequently, limiting the production of pro-inflammatory adipocytokines [299].
**TUG-1197 (compound 34)**	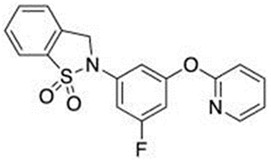 2-(3-fluoro-5-pyridin-2-yloxyphenyl)-3H-1,2-benzothiazole 1,1-dioxide	24 nM [300]	Fully selective for GPR120 and very potent	- Improved glucose tolerance after oral dosing in an oral glucose tolerance test performed both in normal and diet-induced obese (DIO) mice [72,284]. - Chronic dosing of TUG-1197 in DIO mice increased insulin sensitivity in a GPR120-dependent manner [284]. - Improvement in glucose homeostasis correlated with a moderate but significant reduction in bodyweight in DIO mice [284].

* based on Ca^2+^ mobilization (EC50); values for human FFAR4/GPR120 unless noted; ^#^ N.A.D.—denotes no data or ambiguous data.

**Table 3 ijms-26-02501-t003:** General limitations in the therapeutic use of GPR120/FFAR4 agonists in obesity-related metabolic disorders and future strategies to overcome them.

Origin of the GPR120/FFAR4 Agonist	Example Compounds	General Limitations in Therapeutic Use in Obesity-Related Metabolic Disorders	Future Strategies
**NATURAL SOURCES (non-LCFA agonists)**	Grifolin derivatives (grifolic acid and grifolic acid methyl ether), KDT501 (isohumulone), Teadenol A	- The results come almost exclusively from studies on animals and cell cultures [74,233]. - In the case of non-selective agonists, it is difficult to determine which metabolic pathways are responsible for the therapeutic and/or adverse effects via GPR120 and which via other receptors (e.g., GPR40, PPAR-γ) [70,235]. - Most often low potency (partial agonists) [74]. - No clinical data on human dosage and the range between therapeutic and toxic doses (therapeutic index) [74,202]. - There are no complete data on changes in GPR120 expression in specific tissues in obesity and metabolic syndromes [185,301].	- Due to the virtually unlimited possibilities of designing synthetic GPR120 ligands, the chapter on the development of natural GPR120 agonists as future drugs should be treated as closed [77,216].
**SYNTHETIC**	GW9508, NCG21, TUG-891, Compound **29** (phenylpropanoic acid), Compound **18** (chromane propionic acid derivative), Compound A (Merck), Metabolex 36, AZ13581837, GPR120 agonist III, GSK137647A, TUG-1197 (compound **34**)	- The results come almost exclusively from studies on animals and cell cultures [74,266]. - Finding a compromise between the therapeutic efficacy of the active substance of the drug and its pharmacokinetic and pharmacodynamic properties [277,278]. - In the case of non-selective agonists, it is difficult to determine which metabolic pathways are responsible for the therapeutic and/or adverse effects via GPR120 and which via other receptors (e.g., GPR40, PPAR-γ) [70,235]. - No clinical data on human dosage and the range between therapeutic and toxic doses (therapeutic index) [74,202]. - Incomplete knowledge of GPR120 signaling pathways, the selective activation of which will ensure optimal drug action [302]. - There are no complete data on changes in GPR120 expression in specific tissues in obesity and metabolic syndromes [185,301].	- Commencement of large-scale clinical trials using computer-designed molecules with a precisely defined three-dimensional structure, GPR120 agonists, and then selection of those substances that have allosteric-related, particularly strong/selective effects on metabolic pathways disturbed in obesity [77,216,283,302]. - Solving safety issues (side effects and complications of therapy) [72,202]. - Assessment of the impact of individual variability on expected therapeutic effects [72,202,302].

## Data Availability

No new data were created. Instead, the data are quoted from the available cited literature.

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
