# Peer review of "Potential Therapeutic Exploitation of G Protein-Coupled Receptor 120 (GPR120/FFAR4) Signaling in Obesity-Related Metabolic Disorders"

_ijms, 2025, doi:10.3390/ijms26062501_

Round 1
Reviewer 1 Report
Comments and Suggestions for Authors
In this review, Dariusz Szukiewicz explored the role of G-protein coupled receptor 120 (GPR120) in obesity-related metabolic disorders. This review significantly contributes to obesity-related metabolic disorders by highlighting GPR120's role in regulating inflammatory responses and insulin sensitivity, proposing it as a promising target for new therapeutic interventions. The manuscript is well-written, and I have only very minor comments:
- Line 64 “IR”, please define the abbreviation.
- Line 89 “transmembrane heptahelical structure (7TM)” has been defined before.
- Lack of references to the literature in some sentences, such as line 131.
- Line 168, according to reference 41, there are approximately 350 350 non-olfactory members in humans.
- Line 246, reference 215 was not mentioned in the text.
- The order of references needs to be rearranged, such as reference 216 after reference 244.
Reviewer 2 Report
Comments and Suggestions for Authors
This timely review manuscript provides comprehensive information on G Protein-coupled 2
receptor 120 (GPR120/FFAR4) and its role and involvement in anti-inflammatory and metabolic signaling. After a detailed introduction, the main objective of this review is outlined, i.e. giving an overview of the current stage of therapeutic interventions on obesity-related metabolic diseases via targeting GPR120/FFAR4. As the detrimental impact of obesity and related disorders (particularly in Western countries) is significant and still growing, the manuscript is of interest to the readership of IJMS. It will be beneficial for both novices and experinced researchers in the field.
The well-structured review indicates essential difficulties in deciphering the actual role of LCFAs as natural ligands in GPR120 signaling as well (l. 264-272). In direct connection with that, the significance of the dietary context for GPR120 expression is emphasized. However, the review is focussed on natural and (novel) synthetic GPR120 ligands as therapeutic options in obesity-related disorders, but is essentially restricted to those without giving much information about the importance of the dietary context.
As an amendment a concise section should be added, describing the significance of a proper nutrition (i.e. emphasizing the food quality, such as healthy fats, sufficient micronutrients) and avoiding/reducing the uptake of various toxins combined with efficient detoxification protocols, which will have a (synergistic) influence on the efficacy of GPR120 ligands. For example, the benefits of various FAs, such as MCAs in cocos fat whose weight-controlling effects are well known, may be mentioned.
Reviewer 3 Report
Comments and Suggestions for Authors
This review has presented the current state of knowledge concerning the involvement of GPR120 in anti-inflammatory and metabolic signaling, which may provide a rationale for the development of novel, GPR120-based therapies for overweight and obese individuals. The main problems associated with introducing this type of treatment into clinical practice are also discussed. The topic is interesting, with enough latest referneces.It was well written, which provides a comprehensive summary for the research field. The following revision could improve the quality of the paper.
Abstract, the rational/logic of GPR120 in anti-inflammatory and metabolic signaling between therapies for overweight and obese individuals needs to be revised more clear.
What is the current state about the using GPR120 as target for the novel drugs develpment for therapies for obese? What is the limitation for the development/use of it. Please summarize the current situation/data in a table, which will benefit for the readers.
L135-148, it is better to summarize this part as one paragraph.
Table 1 provided the naturally occurring GPR120 agonists with potentially beneficial effects in obesity-related metabolic disorders. What is the limitation about the use of it?
Table 2, the limitation and strategy also should be added in the table.
Concluding Remarks, the authors should highlight the current limitation about the development/use of GPR120-based therapies for overweight and obese individuals.
Please check and make sure all the abbreviation should be used when its full name was firstly appeared in the manuscript.
References, some old, such as 36, if not necessary, which might be removed or replaced.
